# Detection of Single Molecules Using Stochastic Resonance of Bistable Oligomers

**DOI:** 10.3390/nano10122519

**Published:** 2020-12-15

**Authors:** Anastasia Markina, Alexander Muratov, Vladislav Petrovskii, Vladik Avetisov

**Affiliations:** N. N. Semenov Federal Research Center for Chemical Physics Russian Academy of Sciences, Kosygina 4, 119991 Moscow, Russia; ad.muratov@physics.msu.ru (A.M.); petrovskyy@gmail.com (V.P.)

**Keywords:** thermoresponsive oligomers, nanomechanics, bistability, stochastic resonance, single molecules detection

## Abstract

Ultra-sensitive elements for nanoscale devices capable of detecting single molecules are in demand for many important applications. It is generally accepted that the inevitable stochastic disturbance of a sensing element by its surroundings will limit detection at the molecular level. However, a phenomenon exists (stochastic resonance) in which the environmental noise acts abnormally: it amplifies, rather than distorts, a weak signal. Stochastic resonance is inherent in non-linear bistable systems with criticality at which the bistability emerges. Our computer simulations have shown that the large-scale conformational dynamics of a short oligomeric fragment of thermosrespective polymer, poly-*N*-isopropylmethacrylamid, resemble the mechanical movement of nonlinear bistable systems. The oligomers we have studied demonstrate spontaneous vibrations and stochastic resonance activated by conventional thermal noise. We have observed reasonable shifts of the spontaneous vibrations and stochastic resonance modes when attaching an analyte molecule to the oligomer. Our simulations have shown that spontaneous vibrations and stochastic resonance of the bistable thermoresponsive oligomers are sensitive to both the analyte molecular mass and the binding affinity. All these effects indicate that the oligomers with mechanic-like bistability may be utilized as ultrasensitive operational units capable of detecting single molecules.

## 1. Introduction

The desire to build detecting, control, and logic elements as small as possible actively stimulates the search for molecular structures of nanometer size, capable of performing discrete operations at the molecular level. To date, impressive advances have been made in mastering various nanodevices, e.g., switches [1,2], sensors [3,4,5,6], catalytic agents [7], actuators [8], and logic gates [9]. Submicron-sized mechanical and electromechanical machines are also an important advance [10,11,12,13,14]. However, the design of molecular machines a few nanometers in size remains a challenge.

A nanometer-sized molecule must have a specific dynamic in order to work as a machine. Indeed, such a molecule, being large enough, has a huge number of degrees of freedom. However, machine-like action implies low-dimensional dynamics, which are supposed to be realized through collective atomic motions associated with the one or two slowest degrees of freedom of the molecule. To be distinguished dynamically, the functional degrees of freedom must be separated from all faster degrees of freedom by a large spectral gap [15,16,17]. Therefore, the search for molecular structures performing machine-like action is severely limited.

The type of operation can further narrow the scope of the search. Switching, for example, implies an abrupt change in state of a two-state system when the controlling stimulus crosses a threshold value. In terms of dynamical systems, switching is a non-linear dynamic with criticality [18,19].

A commonplace metal ruler subjected to longitudinal compression is a simple demo system with critical behavior. Indeed, the ruler remains straight under light compression. However, as soon as the compressive force exceeds a certain critical value, the straightened state becomes unstable and the ruler bends up or down. Above critical compression, the ruler is a bistable dynamical system and can jump from one state to another. In dynamical terms, the compression transforms one-state dynamics into bistable dynamics, which are characterized by potential energy with two minima separated by a bistability barrier. The ability of a metal plate to undergo abrupt changes in states via applied power loads is exactly why it is used in mechanical switches.

Bistability is interesting not only for the discrete action. If random perturbations can activate transitions over the bistability barrier, the bistable system will spontaneously jump between two states, performing spontaneous vibrations. The time intervals separating spontaneous jumps, i.e., the lifetimes of the system in each of the two states, are widely distributed around a mean value, which, in turn, depends exponentially on the ratio of the bistability barrier to the intensity of the perturbation [20,21]. The exponential dependence enables the transformation of spontaneous vibrations into regular jumps via a slight wiggle of the bistable potential by a weak oscillating force. Regularization of spontaneous vibrations by a weak oscillating force was called stochastic resonance [20], mainly because the output associated with noise-induced transitions of the bistable system can be much larger than the weak oscillating force used as the input. In this sense, one can say that stochastic resonance is a phenomenon involving the amplification of a weak signal by noise. That is why stochastic resonance has attracted much interest, in particular, for sensing [22,23].

In macroscale dynamical systems, the bistability barrier is macroscopically high, so spontaneous vibrations of the macroscopic-sized bistable systems, e.g., of the micron- or even submicron-sized ones, cannot be activated by thermal-bath fluctuations. Much stronger perturbations are needed. However, thermal fluctuations can be major perturbations for systems a few nanometers in size. Therefore, if a nanometer-sized molecule is bistable, and its bistability barrier is comparable to the thermal-bath fluctuations energy, spontaneous vibrations and stochastic resonance will appear naturally. Thermally activated spontaneous vibrations and stochastic resonance seem to be highly attractive for measuring at the molecular level.

Searching for mechanic-like bistability among molecules a few nanometers in size may not seem promising, but computer simulations give some hope [24]. Our intensive molecular dynamic simulations of rather short oligomeric fragments of thermoresponsive polymer poly-*N*-isopropyl-methyl-acrylamide (PNIPMA) subjected to longitudinal compression revealed specific oligomeric samples that successfully combine nanometer size and mechanic-like bistability. These oligomers do exhibit thermally activated spontaneous vibrations and stochastic resonance, and both of these modes of bistability turned out to be sensitive to the attachment of single molecules. In this article, we present the computer simulation data on the mechanic-like bistability of a syndiotactic *N*-isopropylmethylacrylamid oligomer with a length of 30 units (oligo-30s-NIPMAm), as well as the data on the sensitivity of the oligomer spontaneous vibrations and stochastic resonance to the attachment of single molecules.

## 2. Materials and Simulation Method

The simulation approach consists of the following steps: (i) equilibrium conformation of oligo-30s-NIPMAm below and under critical temperature (~32 °C); (ii) application of a compression force to a closed 30s-NIPMAm conformation to observe spontaneous vibrations; (iii) application of external periodic stimuli to a critical compression force to observe stochastic resonance; (iv) addition of an analyte to 30s-NIPMAm to show a sensing regime.

We started with morphology simulations using the GROMACS 2019 simulation package [25]. In our approach, we first adapted the OPLS-AA force-field [26,27,28]. Because all Lennard–Jones parameters were taken from the OPLS-AA, the combination rules and the fudge factor of 0.5 were used for 1–4 interactions. The long-range electrostatic interactions were treated by using a smooth particle mesh Ewald technique. All calculations were performed in the NVT ensemble using the canonical velocity-rescaling thermostat, as implemented in the GROMACS simulation package. The oligo-30s-NIPMAm and the environmental water were modeled in a fully atomistic representation in a box sized 6.0 × 8.0 × 8.0 nm with a time step of 1 fs. The temperature was set at 290 K, i.e.,sufficiently below the low critical solution temperature of the poly-NIPMAm [29,30,31,32].

The second step was a simulation of spontaneous vibrations. For that, we applied a longitudinal load. After the definition of a critical force for observing spontaneous vibration, we applied a weak oscillating force. Computational details of the application of longitudinal load and weak oscillation force for the oligomer in the free and sensing regime are described in Appendix A.

## 3. Results

In this section, we describe a set of specific characteristics of the oligo-30s-NIPMAm dynamics, which unambiguously show that the oligomer, when subjected to power loads, behaves like a nonlinear system with criticality. Mechanic-like bistability, spontaneous vibration, and stochastic resonance are the focus of our simulations. In addition, we present the data corresponding to the sensitivity of the spontaneous vibrational and stochastic resonance modes of the oligomer to the attachment of single molecules.

### 3.1. Oligomeric Templates for Molecular Dynamic Simulations

We started with simulations of the molecular dynamics of oligo-30s-NIPAm at different temperatures to ensure that the oligomer was thermoresponsive, i.e., it actually had two well-defined conformational states (called, hereafter, “open” and “closed” states) and sharply changed the states when the temperature crossed a critical value. The oligomer states were characterized by the distance between the oligomer ends, while the oligomer conformation was additionally controlled by the gyration radius. The low critical solution temperature of the oligo-30s-NIPAm was specified to be close to 305 K [29,30,31,32], so a temperature equal to 290 K was chosen for the preparation of samples in the open state, while it was equal to 320 K for samples in the close state. Figure 1a shows the shapes of the open and closed states of oligo-30s-NIPAm equilibrated at 290 K and 320 K, respectively. The fluctuations of the end-to-end distance, Re, at these two states are shown in Figure 1b.

### 3.2. Bistability and Spontaneous Vibrations

Assuming that a longitudinal compression of the oligomer could lead to a sharp transition from the open state to the closed state as the compression exceeded a critical value, we took the oligomer equilibrated in the open state (at 290 K) and applied a longitudinal compressive force, F, in the same manner as in the demo-ruler above (Figure 2a). In fact, one could take the oligomer equilibrated in the closed state at 320 K and apply the pulling forces. Such experiments also were checked, and the results were qualitatively the same.

Figure 2b shows how the stationary states of the oligomer evolve when the compression grows.

One can clearly see a drastic change in the oligomer dynamics near the critical point F_c_ ≈ 320 pN. Indeed, the open state remains the only state of the oligomer up to the critical compression. However, at the critical compression, a new branch of actually bent stationary states appears. Far from the critical point, the oligomer is found in the open or closed state, depending on compression. If the compressive force quickly crossed the critical point, then the oligomer would abruptly transition from the open state to the closed state, like a jump-like switching of a metal ruler. Somewhat above the critical compression, low-frequency spontaneous vibrations between the open and closed states are observed, i.e., the oligomer is bistable in this region. It should be noted that spontaneous vibrations are not observed just after the critical point, despite the fact that there are two branches of states. Spontaneous vibrations are observed with some offset from the critical point, i.e., in the region where the bistability barrier matches the thermal fluctuations. The bistability barrier should be neither too small nor too large relative to thermal fluctuations.

Spontaneous vibrations are observed with some offset from the critical point, i.e., in the region where the bistability barrier matches the thermal fluctuations. Indeed, as long as the barrier remains less than kT, the thermal noise will blur the states. Therefore, the bistability barrier should be noticeably larger than kT. The bistability barrier should be neither too small nor too large relative to thermal fluctuations in order to observe spontaneous vibrations as random transitions between two well-defined states.

To more accurately verify the spontaneous vibrations, we plotted the probability distributions, P(Re), of the states Re averaged over a set of dynamic trajectories, Re(*t*), and studied how these distributions were transformed when the compression grew. The bifurcation diagram in Figure 2b is reconstructed exactly from these data. Below and fairly above the bifurcation point F_c_ ≈ 320 pN, the oligomer dynamics are characterized by single-peak distributions P(Re) at the open or closed state, respectively. Near the bifurcation point from above, there is an interval of compressions in which the distributions P(Re) have a double-peak form caused by spontaneous vibrations of the oligomer between the open and closed states. In our simulations, the mean lifetime of the states, i.e., the mean value of random time intervals between the jumps defined along dynamic trajectories, ranged from 5 to 10 ns, depending on the compression. Using Kramer’s exponential relation between the lifetime of the states and the bistability barrier, and assuming that the pre-exponent collision factor is equal to 10^−13^ s, the bistability barrier was estimated as 10–15 *k*_B_*T*, where *k*_B_ is the Boltzmann constant and *T* is the bath temperature. Following this estimate, we assumed that the reordering of hydrogen bonds between the oligomer and surrounding water could activate spontaneous vibrations of the oligo-30s-NIPMAm. If this were the case, the spontaneous vibrations would be caused precisely by the thermoresponsibility of the oligo-30s-NIPMAm, e.g., due to the switching of hydrogen bonds from the oligomer–water configuration to the oligomer–oligomer configuration [30,31,32].

To verify whether the mechanic-like spontaneous vibrations of the oligomer were, indeed, controlled by reversible switching of some hydrogen bonds, we checked the hydrogen bonds in the oligomer–water configuration and in the oligomer–oligomer configuration and studied how the number of these hydrogen bonds fluctuated. We realized that only the hydrogen bonds located in the oligomer bending area had a reasonable relation to the oligomer spontaneous vibration. These data are shown in Figure 3a.

One can see that spontaneous vibrations of the oligomer and the switching of particular hydrogen bonds are synchronized on antiphase with high correlation coefficients of −0.90. Figure 3b demonstrates that no significant correlation between the oligomer dynamics and the switching of hydrogen bonds is observed beyond the bistability region.

### 3.3. Stochatic Resonance

Taking the oligomer in the spontaneous vibrations regime, we stimulate the stochastic resonance by applying an additional weak oscillating force directed along the compressive force, F (see Figure 2). The oscillating force was induced by an external oscillating electrical field *E* = *E*_0_cosω*t* enacted on a charge (+1) set at one end of the oligomer, while a compensative charge was set at the opposite end. Following the well-known fact that the main resonance peak arises when the frequency of the oscillating field matches the mean value of the state lifetimes in the spontaneous vibrations mode [23], we tested the oci fields with the period of *T* = 5 ns (200 MGz in frequency); the amplitudes ranged from 0.1 V/nm to 1.0 V/nm. For more details, see Appendix A.

The explicit manifestation of the stochastic resonance induced by the oscillating force with the frequency of 200 MHz and the amplitude of 0.2 V/nm is shown in Figure 4.

### 3.4. Single Molecules Sensing via Spontaneous Vibrations Mode

This series of computer experiments aimed to study the bistable dynamics of the 30s-NIPMAm when a molecular cargo was attached. It is known that nonlinear systems are sensitive to weak impact precisely near the bifurcation point [32,33,34,35]. Therefore, we first initiated the spontaneous vibration of unloaded oligo-30s-NIPMAm near the critical compression and then studied how the oligomer dynamics changed with molecular cargo attachment (see Figure 5a). Figure 5b,c show the response of spontaneous vibrations of the oligo-30s-NIPMAm at the compression of 375 pN to the attachment of a dye molecule (ATTO 390).

Recall that the unloaded oligomer subjected to the same compression is bistable and vibrates spontaneously. However, as Figure 5b shows, the oligo-30s-NIPMAm escapes the spontaneous vibrations mode when a molecular cargo attaches to the oligomer. The loading by a cargo shifts the bistability region. This fact is additionally confirmed by the evolution of statistical weight distributions for visiting the open and closed states when the compressing force is varied. In particular, the attachment of a dye molecule shifts the compression under which the spontaneous vibrations are observed from 375 pN to 390 pN (Figure 5c). Note that a molecular cargo shifts the spontaneous vibrations mode toward the higher compression of the oligomer. The shifts must depend on the type and number of the molecules attached. The shifts of the spontaneous vibrations mode caused by the attachment of various molecular cargos are presented in Appendix A.

### 3.5. Single Molecules Sensing via Stochastic Resonance

Taking the oligomer in the spontaneous vibrations mode at the compression of 375 pN, we first initiated the stochastic resonance mode by applying an additional oscillating field $E=E_{0}cos\omega t$ with the amplitude $E_{0}=0.2$ V/nm and frequency ω=2π/5 ns^−1^, and then studied how the oligomer dynamics changed with a molecular cargo attachment.

The spontaneous vibrations mode (red curve) and the stochastic resonance mode (black curve) for the oligomer without an analyte are shown in Figure 6a.

In the spontaneous resonance mode, the oligomer itself responded to the oscillating force with the resonance frequency. The oligomer dynamics loaded by a molecular cargo are shown in Figure 6b. One can see that, when the cargo is attached, the oligomer completely leaves the spontaneous vibrations mode (Figure 6b). The stochastic resonance mode is lost too, but the oligomer still vibrates, though the vibrations are not well-synchronized with the external harmonic field. These experiments showed that the attachment of a molecular cargo transformed the stochastic resonance mode into irregular jumps characteristic of spontaneous vibrations. We have tested various types of cargos, as well as various numbers of attached cargo-molecules (one, two, or three). This allows us to study the effect of the mass of the analyte on oscillation characteristics (see the Appendix A).

To see this transformation more clearly, we slightly increased the compressive force up to 390 pN and first generated the stochastic resonance of the oligo-30s-NIPMAm without a molecular cargo by applying the harmonic field of the amplitude $E_{0}=0.2\;$ V/nm and frequency ω=2π/5  ns^−1^. Then we attached an analyte and generated new trajectories. The analyte-induced transformation of the stochastic resonance mode is clearly seen in a comparison of these two sets of trajectories, shown in Figure 7a,d. Figure 7b,e show autocorrelation functions. One can see that upon attaching the analyte molecule, the autocorrelation function is blurred. This fact is also supported by the furrier spectrum of these autocorrelation functions (Figure 7c,f), where the main peak is significantly lower for the system with the analyte.

Summarizing this set of computer experiments, we conclude that the attachment of a molecular cargo shifts the stochastic resonance mode and its characteristics. This shift depends on various factors, such as binding motive and the mass of the analyte. The stochastic resonance mode can blur, or the resonance can transform into spontaneous vibrations, or can leave the vibrational mode. It is important that the spontaneous vibrations and stochastic resonance modes are sensitive to both the analyte’s molecular mass and its binding affinity (for more details, see Appendix A).

## 4. Discussion

In most sensing applications, it is generally accepted that the inevitable stochastic disturbance imparted by the surroundings limits the detection signal at the molecular level. However, noise can be leveraged to amplify, rather than distort, a weak signal—a phenomenon known as stochastic resonance. Our computer simulation studies of nanometer-sized oligomeric fragments of thermoresponsive polymer poly-*N*-isopropyl-methyl-acrylamide subjected to longitudinal compression have shown that the large-scale conformational dynamic of the oligomers can exhibit non-linear dynamics similar to the dynamics of bistable mechanical systems. To manifest such behavior, the oligomer should be about two Kuhn length in size. Aside from the spontaneous vibrations and stochastic resonance, which in our case are activated by conventional thermal noise, these dynamic modes turn out to be sensitive to single molecule attachment. When a molecular cargo attaches, either the stochastic resonance mode is blurred, or the resonance is transformed into the spontaneous vibrations mode, or the oligomer completely exits the vibrational mode. It is important that the spontaneous vibrations and stochastic resonance of the bistable oligomers are sensitive to both the analyte’s molecular mass and its binding affinity (for more details, see Appendix A). All these effects manifest themselves to a sufficiently significant extent to be used in detecting the masses of single molecules in solutions. Therefore, bistable oligomers can act as ultrasensitive materials capable of detecting single molecules against the background of natural stochasticity at the molecular level. Due to the small size of the oligomers, the stochastic resonance visibly shifts even when the attached molecule is about 200 Dalton in mass. This is sufficient for the detection of single molecules, e.g., hormones we have used as an analyte in our studies.

Importantly, the oligo-30s-NIPMAm that we established in this article is not a unique sample. Several oligomers of thermosensitive polymers have properties suited for exhibiting nano-mechanical bistability and detecting small molecules.

## Figures and Tables

**Figure 1 nanomaterials-10-02519-f001:**
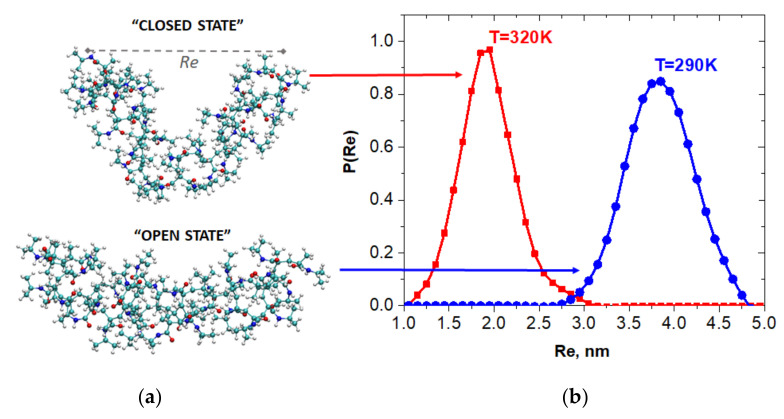
Temperature-induced bistability of the oligo-30s-NIPMAm. (**a**) Typical shapes of the oligomer in the “open” and “closed” states equilibrated at the temperatures 90 K and T = 320 K, respectively. (**b**) Normalized distributions of fluctuations of the end-to-end distances Re at the open and closed states.

**Figure 2 nanomaterials-10-02519-f002:**
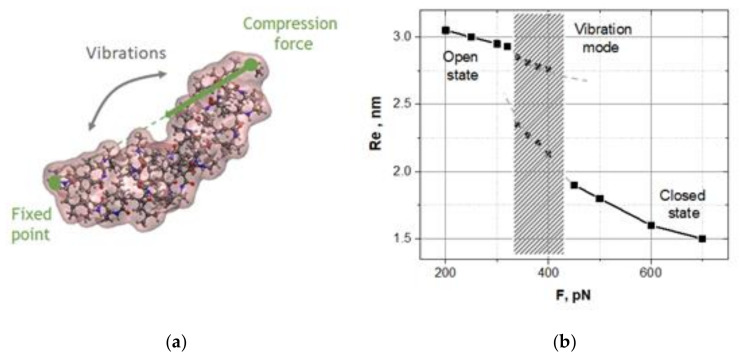
The response of the oligo-30s-NIPMAm to longitudinal compressions: (**a**) Schematic presentation of the oligomer compression; (**b**) The bifurcation diagram represents how the stationary states, Re, of the oligomer depend on compression F as a control parameter. A shadow interval of compression forces from 340–430 pN marks the area of spontaneous vibrations.

**Figure 3 nanomaterials-10-02519-f003:**
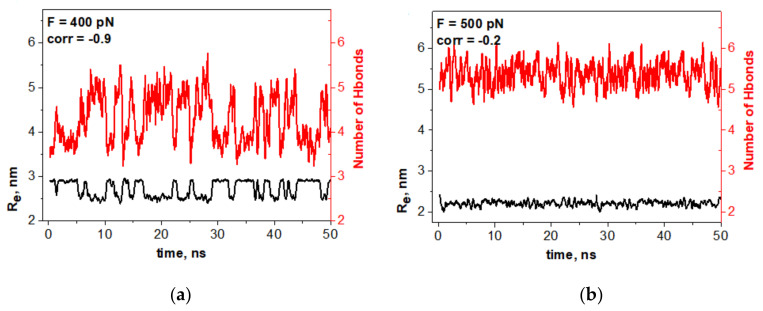
Correlation between the oligomer dynamics R_e_(*t*) and switching of hydrogen bonds located in the oligomer bending area. The compressions *F* are indicated on the panels. (**a**) Spontaneous vibrations of the oligomer (black curves, left axes, Re nm) strongly correlate with the switching of hydrogen bonds from the oligomer–oligomer configuration to the oligomer–water configuration and back (red curves; right axes). (**b**) No significant correlation between the oligomer dynamics and the fluctuations of hydrogen bonds is seen beyond the bistability region.

**Figure 4 nanomaterials-10-02519-f004:**
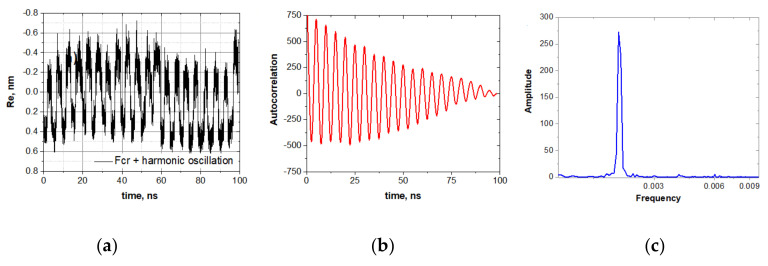
Stochastic resonance generated by harmonic electric field *E* = *E*_0_cosω*t* with the amplitude 0.2 V/nm and the frequency 200 MGz: (**a**) The oligomer dynamic Re(*t*) in the stochastic resonance mode; (**b**) The autocorrelation function of Re(*t*); (**c**) The frequency spectrum of the autocorrelation function shown on panel (**b**).

**Figure 5 nanomaterials-10-02519-f005:**
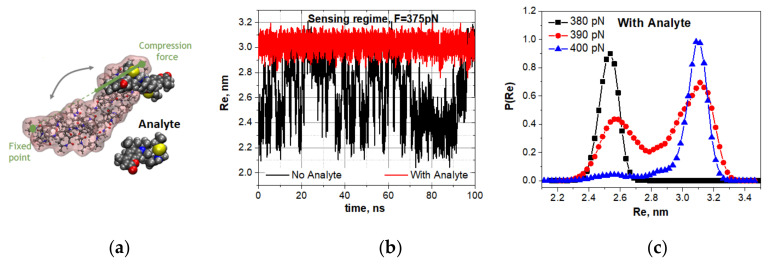
Sensitivity of the oligo-30s-NIPMAm spontaneous vibrations to the attachment of a single-molecule cargo (an analyte). (**a**) Schematic presentation of the computer experiments. (**b**) Spontaneous vibrations of the unloaded oligomer (black trajectory) at the compression of 375 pN, and non-vibrating dynamics of the oligomer loaded with an analyte at the same compressing force (rad trajectory). (**c**) Evolution of statistical weights for visiting the open and closed states by the loaded oligomer vs. the compressive force. Molecular cargo (an analyte) shifts the spontaneous vibrations mode from the compression of 375 pN (see panel (**b**)) to the compressive of 390 pN.

**Figure 6 nanomaterials-10-02519-f006:**
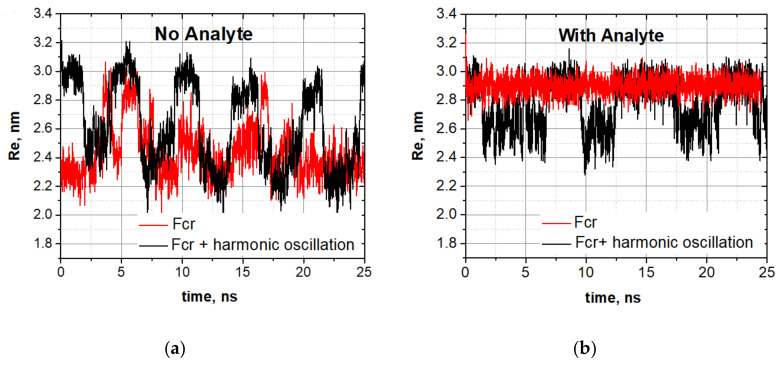
Stochastic resonance and the sensing regime. (**a**) Spontaneous vibrations (red curve) and stochastic resonance (black curve) of the unloaded oligo-30s-NIPMAm under the compression of 375 pN. (**b**) Dynamics of the oligomer loaded by a tryptophan molecule at the same conditions. Trajectories are marked as in panel (**a**).

**Figure 7 nanomaterials-10-02519-f007:**
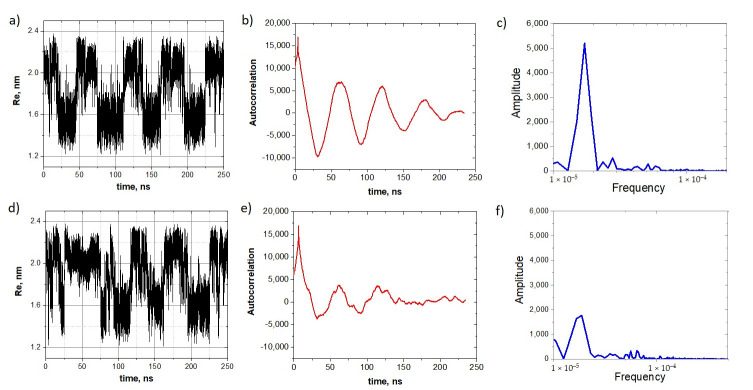
The analyte-induced transformation of the stochastic resonance mode. Vibrations of the oligo-30s-NIPMAm before (top row of panels) and after (bottom row of panels) the attachment of a tryptophan molecule: (**a**–**c**) Stochastic resonance mode of the oligo-30s-NIPMAm, its autocorrelation function, and the frequency spectrum. (**d**–**f**) Distortion of the spontaneous resonance mode caused by the attachment of a molecular cargo to the oligomer.

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
