# Peer review of "Detection of Single Molecules Using Stochastic Resonance of Bistable Oligomers"

_nanomaterials, 2020, doi:10.3390/nano10122519_

Round 1
Reviewer 1 Report
In this paper, the authors use molecular dynamics simulations to study the transition of a small oligomer (oligo-30s-NIPMAm) between two bistable structural forms (open/closed). In particular they explore the resonant switching between the two states when the oligomer is subjected to an oscillating force and the impact of binding of molecules on this resonance.
The paper is globally well written and quite easy to follow. The conclusions of the authors are most of the time in line with the data. Finally, the numerical details given seem sufficient to be able to reproduce the work if needed. I have only few comments and I think this paper could be published but with major revisions.
My first comment concerns the protocol that the authors have put in place in order to add their oscillatory force on the system to induce the stochastic resonance. To do so, they placed two extra charges at both ends of the oligomer and applied an external electrical field. I wonder why they did not chose instead to add an oscillating component to the compression force that they applied on the molecule. Is this a technical issue? This may not change too much the results, but applying the electric field, every partial atomic charge should respond, and so the whole dynamics of the system could be modified. Unless the electric field is only “seen” by the extra charges (but it is not made clear in the paper)? I think it would be nice to ensure that this ad-hoc protocol does not induce unwanted effects.
My other major comment concerns the conclusions about the role of the mass and the binding affinity of the molecular cargo on the stochastic resonance. First of all I am surprised that the authors don’t show results about this aspect in the main text, whereas this seem to be an important issue for them (it is cited in the introduction and in the discussion section). I really think that if this issue is important, the results should be made more explicit. However, with the only 3 molecular cargos that the authors have studied, I am not sure that any conclusion regarding the role of the mass and/or the binding affinity to the oligomer can be drawn. To my point of view, all that can be said is that the effect is sensitive to the molecular cargo. In order to be more specific, a more thorough study should be conducted using more cargos, some of which having the same mass, the same “shape”, the same binding affinity to the oligomer, etc. so that it could be possible to study the impact of each parameter.
Another comment concerns the end of the results section and especially the discussion of figure 7 that is very “weak”. I think the authors should comment a bit more the results of this figure. What is more, in Figure 7 (d-f), what is the molecular cargo attached to the oligomer?
The last sentence of the results section (“Summarizing this set of computer experiments, we conclude that the attachment of a molecular cargo can manifest itself in different ways) is also very elliptic. I think the authors could be more specific about the “different ways”.
Author Response
Comments of Reviewer #1, Author Responses, and Manuscript Revisions
We thank Reviewer#1 for his or her time and effort in reviewing our manuscript. We hope that our comments will address main the reviewer’s concerns.
Comment 1: “My first comment concerns the protocol that the authors have put in place in order to add their oscillatory force on the system to induce the stochastic resonance. To do so, they placed two extra charges at both ends of the oligomer and applied an external electrical field. I wonder why they did not choose instead to add an oscillating component to the compression force that they applied on the molecule. Is this a technical issue? This may not change too much the results, but applying the electric field, every partial atomic charge should respond, and so the whole dynamics of the system could be modified. Unless the electric field is only “seen” by the extra charges (but it is not made clear in the paper)? I think it would be nice to ensure that this ad-hoc protocol does not induce unwanted effects.”
Response: Indeed, applying the harmonic electrical field in order to reproduce the oscillating component of the force is connected with technical issues. In the original version of GROMACS software, there is no external oscillating force. We were also concerned about modifications of dynamics that can be related to the introduction of an external electrical field. To check that we repeated the representative part of experiments for two cases: in the presence of additional charges and no electrical field, and without additional charges and in the presence of the electrical field. All results were coherent and didn’t show much difference. We have also implemented oscillating force in our own in-house version of GROMACS to check the difference with the external electrical field. We still didn’t find any significant difference. In order to make our results coherent with previous work, we decided to keep this ad-hoc protocol which involves an external electrical field.
Correction: Thus, in order to avoid further misunderstandings about the external electrical field we added the following part into the Supplementary Materials, line 51-57: “Implementation of an external oscillating electrical field is connected with limitations of GROMACS package. We checked whether the external electrical field modifies the dynamics. To check that we repeat the representative part of experiments for two cases: in the presence of additional charges and no electrical field, and without additional charges and in the presence of the electrical field. All results were coherent and didn’t show much difference. We have also implemented oscillating force in our own in-house version of GROMACS to check the difference with the external electrical field. We still didn’t find any significant difference. ”.
Comment 2: “My other major comment concerns the conclusions about the role of the mass and the binding affinity of the molecular cargo on the stochastic resonance. First of all, I am surprised that the authors don’t show results about this aspect in the main text, whereas this seems to be an important issue for them (it is cited in the introduction and in the discussion section). I really think that if this issue is important, the results should be made more explicit. However, with the only 3 molecular cargos that the authors have studied, I am not sure that any conclusion regarding the role of the mass and/or the binding affinity to the oligomer can be drawn. To my point of view, all that can be said is that the effect is sensitive to molecular cargo. In order to be more specific, a more thorough study should be conducted using more cargos, some of which having the same mass, the same “shape”, the same binding affinity to the oligomer, etc. so that it could be possible to study the impact of each parameter.”
Response and Correction: We agree with the reviewer that three compounds are not enough to make a conclusion regarding the influence of binding affinity. However, we don’t suggest any dependence, we just specified that sensitivity can be affected by biding motive. In order to find dependence on mass, we studied an oligo-NIPAm molecule loaded with one, two, or three molecules of each compound. Therefore, we can see that with the growth of mass of molecular cargo we observe a larger shift in oscillation characteristics, see Figure S6 in SM. In this case, we also don't suggest any law, just notice the correlation. To highlight this part we have added, line 259-262: “We have tested various types of cargos, as well as the various number of attached cargo-molecules (one, two, or three). This allows us to study the effect of the mass of the analyte on oscillation characteristics (see the Supplementary Materials, S3).”
Comment 3: “Another comment concerns the end of the results section and especially the discussion of figure 7 that is very “weak”. I think the authors should comment a bit more the results of this figure. What is more, in Figure 7 (d-f), what is the molecular cargo attached to the oligomer?”
Response and Correction: We have added the following description of Figure 7 in lines 272-277: “The analyte-induced transformation of the stochastic resonance mode is clearly seen in a comparison of these two sets of trajectories, shown in Figures 7 a, d. Figures 7 b, e show autocorrelation functions. One can see that upon attaching the analyte molecule the autocorrelation function is blurring. This fact is also supported by the furrier spectrum of these autocorrelation functions (Figures 7 c, f), the main peak is significantly lower for the system with the analyte.”
Comment 4: “The last sentence of the results section (“Summarizing this set of computer experiments, we conclude that the attachment of a molecular cargo can manifest itself in different ways) is also very elliptic. I think the authors could be more specific about “different ways”.”
Response and Correction: We have changed the specified above sentence accordingly: “Summarizing this set of computer experiments, we conclude that the attachment of a molecular cargo shift stochastic resonance model and its characteristics. This shift depends on various factors such as binding motive and mass of the analyte. The stochastic resonance mode can blur, or the resonance can transform into spontaneous vibrations or can leave the vibrational mode. It is important that the spontaneous vibrations and stochastic resonance modes are sensitive to both the analyte’s molecular mass and its binding affinity (for more details, see Supplementary Materials, Note S3). ”
Reviewer 2 Report
Review attached.

Author Response
Comments of Reviewer #2, Author Responses, and Manuscript Revisions
We thank Reviewer#2 for his or her time and effort in reviewing our manuscript. We hope that our comments will address main the reviewer’s concerns.
Comment 1: “In Line 154, the authors mentioned the necessary “offset” from the critical point for the spontaneous vibration. It would be nicer if a more detailed explanation is given.”
Response and Correction:
Exactly at the critical point, we cannot specify two local minima. If we move away from the critical point, then the bistability barrier will grow and bistability will manifest itself more clearly. However, as long as the barrier remains less than kT, the states will be blurred by the noise. Therefore, the bistability barrier should be noticeably larger than kT in order to observe spontaneous vibrations as random transitions between two well-defined states. This is the physical meaning of "offset".
To give more explanation, we modified the text in Lines 159-164 as follows:
"Spontaneous vibrations are observed with some offset from the critical point, i.e., in the region where the bistability barrier matches the thermal fluctuations. Indeed, as long as the barrier remains less than kBT, the thermal noise will blur the states. Therefore, the bistability barrier should be noticeably larger than kT. The bistability barrier should be neither too small nor too large relative to thermal fluctuations in order to observe spontaneous vibrations as random transitions between two well-defined states."
Comment 2: “As for the discussion of hydrogen bonds in Line 169 and Figure 3, the number of hydrogen bonds changes between four and five. How the water molecules situated at the oligomer bending area? The simulation can answer to this long-lasting question. See also the experimental result regarding PNIPAM stretching and references herein (refs.7-10). https://doi.org/10.1002/macp.201700394”
Response: Indeed, this is an interesting question that we left outside the scope of the current research. We haven’t studied the exact structure of the hydrogen bond network. We used a number of hydrogen bonds to show that switching between open and closed states is due to hydrogen bond switching. The article suggested by the reviewer is a good addition to the exciting literature overview.
Comment 3: “If any comment on molecular length dependence of what they found in this research is made, experimentalists will put more interest in the contents.”
Response and Correction: We have studied various molecular lengths. It is a very fine tune here since we are looking for quite rigid systems with two Kuhn lengths in size. Therefore, this specific length is good for observing oscillation mode for this particular case of oligo-NIPAm. In the case of other thermoresponsive oligomers, one needs to do an additional round of length-conformation dependence. We have added a note in lines 301-302.
Round 2
Reviewer 1 Report
The authors have addressed satisfactorily all the comments of the referees.